



# Phase-controlling the motion of floating wind turbines to reduce wake interactions

Daniel van den Berg[1], Daan van der Hoek[1], Delphine De Tavernier[2], Jonas Gutknecht[1], and Jan-Willem van Wingerden[1]

[1]Delft Center for Systems and Control, Delft University of Technology, Mekelweg 2, 2628 CD Delft, The Netherlands
[2]Wind Energy Section, Delft University of Technology, Kluyverweg 1, 2629 HS Delft, The Netherlands

**Correspondence:** Daan van der Hoek (d.c.vanderhoek@tudelft.nl)

**Abstract.** The wake interaction between wind turbines causes significant losses in wind farm efficiency that can potentially be alleviated using wake control techniques. We provide detailed experimental evidence on how the coupling between the so-called Helix wake control technique and a floating turbine's yaw dynamics can be used to increase wake recovery. Using tomographic particle image velocimetry during wind tunnel experiments, we analysed the wake dynamics and its coupling to a floating wind turbine. The measurements show that ensuring the floating turbine's yaw motion is in phase with the blade pitch dynamics of the Helix technique enables an increase of 12 percentage points in available energy in the flow on top of the Helix method applied to bottom-fixed turbines. We find that the in-phase scenario results in an earlier interaction between tip and hub vortex inside the wake, which leads to the desired breakdown of it, thus accelerating the energy advection into the wake.

## 1 Introduction

Wind energy plays a key role in efforts to decarbonise global energy production. For example, the European Commission aims to increase its offshore wind production from 38 GW today to 450 GW by 2050, in order to meet 30 % of Europe's electricity demand at that time (Costanzo et al., 2022). Meeting this target requires a major expansion of the wind energy production capacity at offshore locations, where the majority of Europe's wind energy resources can be found (Fraile et al., 2021). However, 60 % of these energy resources are located in waters too deep for conventional bottom-fixed wind turbines to be economically feasible (Fraile et al., 2021). It is therefore expected that floating wind turbines will be deployed in wind farms of similar sizes as currently seen with bottom-fixed turbines (WindEurope, 2023). Although individual turbines are capable of operating close to their theoretical maximum efficiency, a wind farm can experience an efficiency drop of up to 40 % (Bastankhah and Porté-Agel, 2016; Howland et al., 2019; Barthelmie et al., 2010).

As a wind turbine extracts energy from the incoming airflow, it leaves a zone of turbulent air with reduced wind speed behind it, referred to as the wake. To mitigate turbine-to-turbine wake interaction, methods such as wake steering (Fleming et al., 2017; Becker et al., 2022; Howland et al., 2022; Stanley et al., 2023), static induction control (van der Hoek et al., 2019; Bossanyi et al., 2022), and dynamic blade pitch control (Goit and Meyers, 2015; Frederik et al., 2020a) have been developed. So far, the development of control approaches to mitigate wake interactions has focused on bottom-fixed wind farms (Meyers et al.,



2022; Houck, 2022). When implementing controllers designed for bottom-fixed turbines on floating turbines, the coupling to the dynamics from the additional six degrees of freedom can significantly affect controller performance (Veen et al., 2012; Stockhouse et al., 2023; Lozon et al., 2024). Furthermore, research into the impact of certain specific (floating) turbine movements on wake stability has garnered increasing interest with results indicating that these can enhance wake mixing (Wei et al., 2024; Mühle et al., 2024; Fontanella et al., 2025; Messmer et al., 2025). Recent studies (van den Berg et al., 2022, 2023, 2024b) revealed that collective and individual pitch control techniques can excite the motion of a floating turbine. The magnitude of the motion is dependent on the excitation frequency of the wake mixing technique and its coupling to the floating turbine dynamics. In the case of collective pitch control, the time-varying magnitude of the thrust force creates a fore-aft motion of the turbine rotor. The coupling between the blade pitch input and this motion was found to reduce the effectiveness of the wake-mixing technique, leading to reduced wake recovery (van den Berg et al., 2023; van den Broek et al., 2023).

In this work, we focus on dynamic individual pitch control, often referred to as the Helix method (Frederik et al., 2020b; van Wingerden et al., 2020). The Helix method is a wake mixing method whereby the turbine blades are pitched such that a helical structure of low wind speed is created in the wake behind the turbine. When applied at the right frequency, wake recovery is significantly accelerated when using the Helix method (van der Hoek et al., 2024). The difference between the wake of an unactuated and actuated turbine can be seen in Fig. 1, which shows two wakes behind the IEA 15MW turbine (Gaertner et al., 2020b) mounted on the VolturnUS-S foundation (Allen et al., 2020).

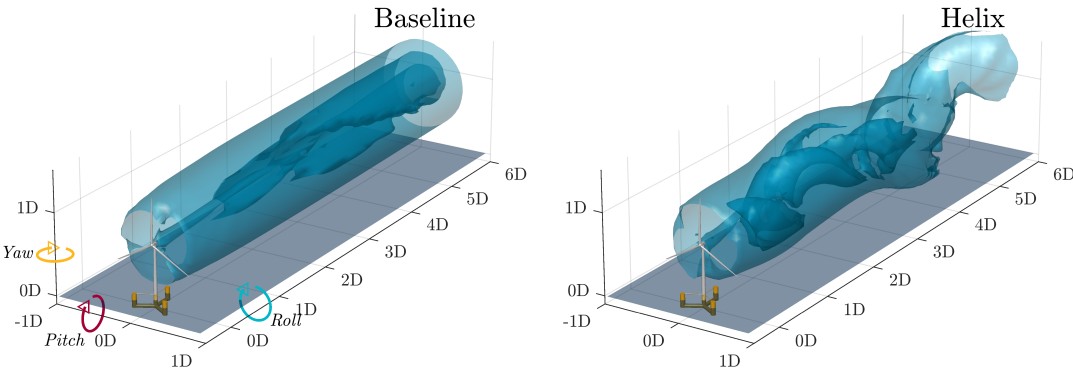

**Figure 1.** Model of the IEA 15MW turbine mounted on the VolturnUS-S floater, with the left figure showing the wake when using a baseline controller, and on the right figure the distorted wake when the Helix method is enabled.

Floating turbines mounted on a semi-submersible foundation like the VolturnUS-S were found to have a natural frequency in the yaw motion near the actuation frequency of the Helix method, resulting in significant yaw motion when excited near or at that frequency (van den Berg et al., 2022, 2024a, b). The frequency at which the Helix is applied is typically characterised by the Strouhal number

$$St = \frac{f_e D}{U_\infty},\tag{1}$$



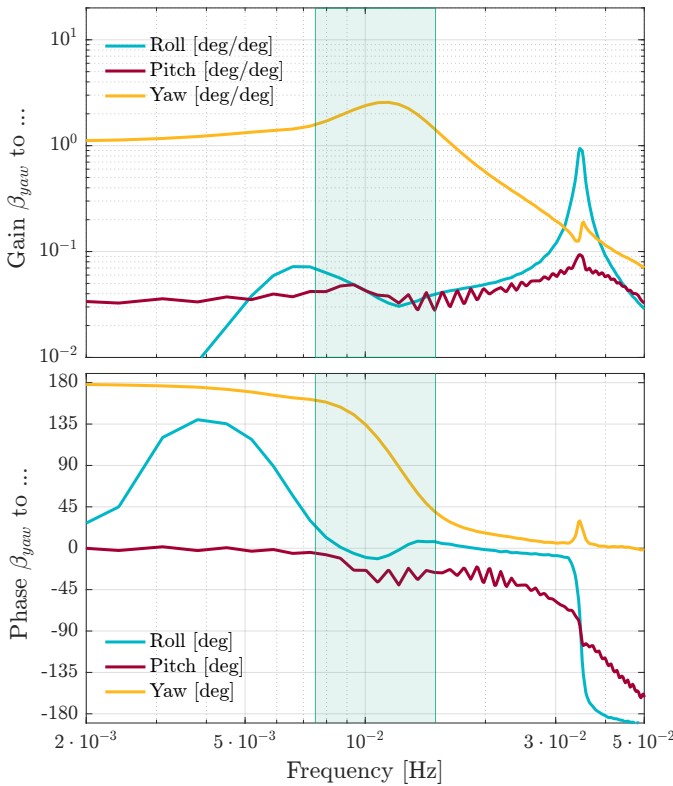

**Figure 2.** Turbine platform's roll (blue line), pitch (red line) and yaw (yellow line) are characterised by their magnitude (top) and phase shift (bottom) with respect to blade pitch input, as a function of Helix excitation frequency. The green shaded area indicates the frequency range $St$ = [0.20, 0.40].

where $f_e$ is the actuation frequency in Hertz, $D$ is the rotor diameter in metres, and $U_\infty$ is the free stream wind speed in metres per second. The frequency at which the Helix is most effective is found to be consistent for different-sized turbines and lies between $St = 0.20$ and $St = 0.40$ when considering two fully aligned turbines spaced a distance of 5 rotor diameters (often referred to as '5$D$') apart (Coquelet, 2022; van der Hoek et al., 2024; Mühle et al., 2024).

The dynamics of a typical floating turbine using the Helix control method are shown in Fig. 2 as a frequency response curve that quantifies the roll, pitch, and yaw angle achieved by applying the Helix method with a blade pitch amplitude of 1 degree. These results are obtained through identification experiments using a full-scale floating turbine in representative sea and weather conditions and QBlade as a simulation suite (Marten, 2020). Appendix A provides greater detail on the experiments carried out for system identification and compares the dynamics of different floating turbines. At its eigenfrequency of approximately $St = 0.30$, every degree of blade pitch results in approximately $2.5°$ yaw angle offset with respect to the incoming wind. A typical $4°$ blade pitch angle input would therefore lead to turbine yaw angles similar to those used for wake steering, albeit in a time-varying fashion. However, due to this specific interaction between the Helix method and the design of a floating turbine, a small change in actuation frequency close to its eigenfrequency greatly affects the phase of the yaw motion, which





impacts the wake mixing performance (van den Berg et al., 2024a). Furthermore, although turbine actuation and movement require some energy, both van den Berg et al. (2024a) and van den Berg et al. (2024b) found that the power generation of a

hypothetical wind farm increases when the floating turbine yaws with the applied Helix method.

This work presents two main contributions investigating this interaction: 1) We describe an experimental setup to study the three-dimensional wake aerodynamics behind a floating turbine, and 2) we show that a change in phase shift between a floating turbine's yaw motion and the Helix method can lead to a reduction or improvement in wake mixing effectiveness of the Helix method.

## 2 Wind Tunnel Experiments


In this section, the mathematical background behind the Helix method is introduced after which the wind tunnel experiments are described in greater detail. The wind tunnel experiments are carried out that combine a hardware-in-the-loop setup and tomographic particle image velocimetry (PIV) to represent and measure the turbine dynamics, the coupled hydrodynamics, and the resulting aerodynamics. The floating turbine is represented using a scaled turbine (Schottler et al., 2016), capable of

applying the Helix method, which is mounted on a hexapod. The yaw motion is imposed on the hexapod, with the imposed motion being representative of an actual floating turbine applying the Helix method. The scaled inputs are derived from the dynamics shown in Fig. 2. The impact of the yaw motion at different phase offsets is quantified by analysing the wind speed in the wake. Tomographic PIV using neutrally buoyant helium-filled soap bubbles (HFSBs) is used to visualise the wake.

### 2.1 The Helix Wake Mixing Method

The Helix wake mixing method is applied in an open loop control scheme by setting sinusoidal input signals to the fixed-frame blade pitch angles. Using the multi-blade-coordinate transformation (MBC) (Bir, 2008) these are transformed into a time-varying individual blade pitch signal that gets applied to the turbine:

$$\begin{bmatrix} \beta_{col}(t) \\ \beta_{tilt}(t) \\ \beta_{yaw}(t) \end{bmatrix} = \frac{2}{3} \begin{bmatrix} 0.5 & 0.5 & 0.5 \\ \cos(\psi_1(t)) & \cos(\psi_2(t)) & \cos(\psi_3(t)) \\ \sin(\psi_1(t)) & \sin(\psi_2(t)) & \sin(\psi_3(t)) \end{bmatrix} \begin{bmatrix} \beta_1(t) \\ \beta_2(t) \\ \beta_3(t) \end{bmatrix}, \tag{2}$$

where $\psi_i$ is the azimuth angle of the blade, and $\beta_{col}$, $\beta_{tilt}$ and $\beta_{yaw}$ are the fixed-frame pitch angles with the subscript *col*

referring to the mean pitch angle of all three blades. The time-varying pitch angles create time-varying out-of-plane bending moments $M_{y,i}$, which can be transformed back into fixed frame moments using the inverse MBC transformation:

$$\begin{bmatrix} M_{y,1}(t) \\ M_{y,2}(t) \\ M_{y,3}(t) \end{bmatrix} = \begin{bmatrix} 1 & \cos(\psi_1(t)) & \sin(\psi_1(t)) \\ 1 & \cos(\psi_2(t)) & \sin(\psi_2(t)) \\ 1 & \cos(\psi_3(t)) & \sin(\psi_3(t)) \end{bmatrix} \begin{bmatrix} M_{col}(t) \\ M_{tilt}(t) \\ M_{yaw}(t) \end{bmatrix}, \tag{3}$$

where $M_{col}$, $M_{tilt}$ and $M_{yaw}$ are the fixed-frame moments with the subscript *col* referring to the collective moment of the turbine. With the Helix method, $M_{tilt}$ and $M_{yaw}$ are varied in a sinusoidal manner, with one signal being phase-shifted by 90°





with respect to the other. These moments are a direct result of the thrust vector being moved off-centre and in a circular motion over the rotor plane when the individual blades are pitched. This also leads to the characteristic helical shape in the wake when this method is applied.

## 2.2 Experimental setup

The experiments were carried out at the Open Jet Facility of the Delft University of Technology, which is an open jet,
closed-circuit wind tunnel with a width and height of 2.85 m. All experiments were run at a constant wind tunnel velocity of $U_\infty = 5$ m s$^{-1}$. The turbulence intensity inside the jet was within the range of 0.5 to 2 %, which was primarily due to the presence of the PIV seeding rake that ejects the helium soap bubbles into the flow (van der Hoek et al., 2024). A modified version of the MoWiTO-0.6 turbine (Schottler et al., 2016; van der Hoek et al., 2024) with a rotor diameter of $D = 0.58$ m was used. The turbine rotor was placed at a safe distance from the turbulent boundary layer of the jet as investigated in (Lignarolo
et al., 2014). Both turbine and hexapod were connected to a dSpace MicroLabBox, enabling real-time control and data transferral between the turbine and hexapod at a sampling rate of $f = 2$ kHz. Once the hexapod was calibrated and zeroed, each of the six degrees of freedom could be controlled and synchronised to the blade pitch input of the wind turbine.

Figure 3 shows the experimental setup. The wake behind the turbine was visualised by neutrally buoyant helium-filled soap bubbles (HFSBs) (Scarano et al., 2015) which were ejected into the flow by a seeding rake with dimensions of 2 m by 1 m.
The HFSBs were illuminated from the side using two LaVision LEDs, enabling the HFSBs to be used as tracers for flow reconstruction. Four Photron FASTCAM SA1.1 high-speed cameras were used to record the wake at 500 frames per second at a resolution of 1024 × 1024 pixels. A multi-axis linear actuator moved the PIV setup downstream of the turbine to measure multiple sections of the wake.

For the experiments, the optimal power coefficient was determined empirically as $C_p = 0.20$. For every experiment, the pitch
angle around which the Helix method was implemented corresponds to the pitch angle of the maximum power coefficient. The turbine was controlled using a PI controller on the generator torque to control rotor speed. With $U_\infty = 5$ m s$^{-1}$, we adjusted the rotor speed $f_r$ such that the optimal tip-speed-ratio $\lambda = \omega_r D/(2U_\infty) = 5$ was achieved, with $\omega_r$ the rotor speed in radians per second. This yielded $f_r \approx 13.7$ revolutions per second.

## 2.3 PIV Data Acquisition

Each PIV measurement consisted of 10 seconds of raw camera footage. The flow tracers were reconstructed using the Shake-The-Box algorithm (Schanz et al., 2016) with Lavision's DAVIS 10 software. On average, each frame consisted of 10,000 reconstructed particles within the measurement volume. After the particle reconstruction, a dataset for all time steps of three-dimensional particle positions and velocities is obtained. For the wake analysis, the particles were spatially averaged to a Cartesian grid over smaller sub-volumes with a Gaussian weighing function. This step entails averaging the velocity infor-
mation of every particle that falls into a sub-volume and assigning those average velocities to that sub-volume for that time step.



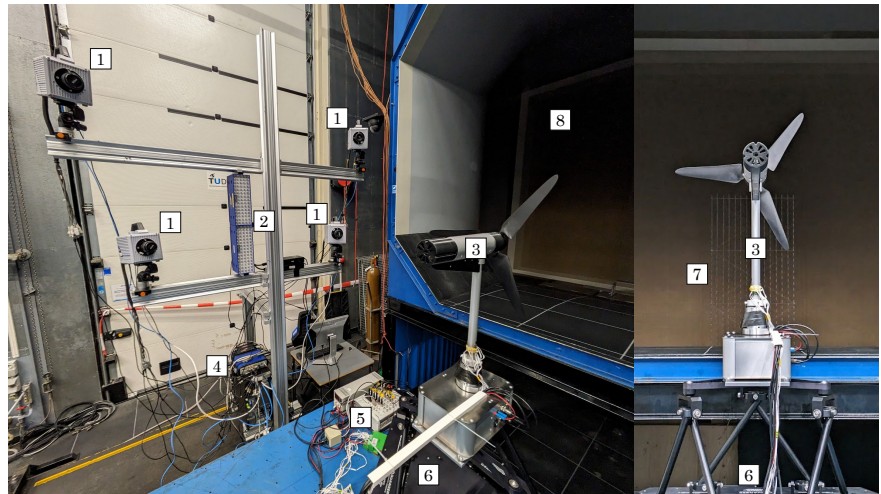

(a)

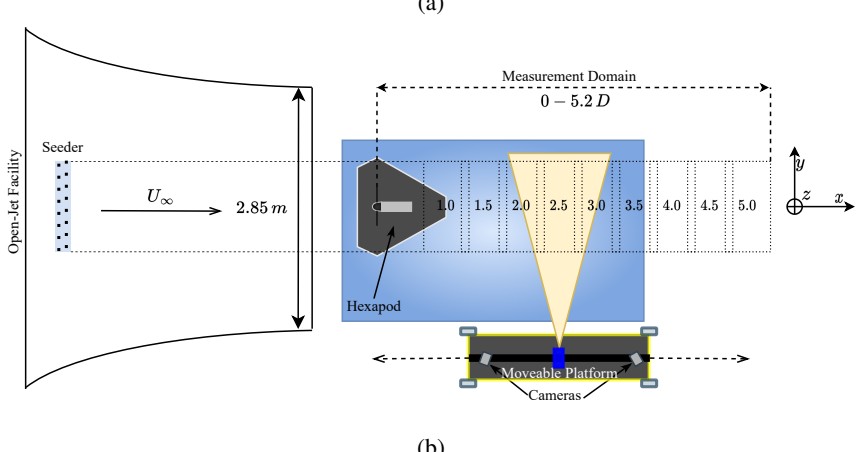

(b)

**Figure 3.** (a) PIV setup consisting of $\boxed{1}$ four Photron FASTCAM SA1.1 high-speed cameras, $\boxed{2}$ two LaVision LED lights used to illuminate the HFSBs, $\boxed{3}$ the MoWiTO-0.6 turbine, $\boxed{4}$ a LaVision PTU-X timing unit used to synchronise the four cameras and LEDs, $\boxed{5}$ dSpace MicroLabBox used for control and data acquisition, $\boxed{6}$ the Quansar Hexapod, $\boxed{7}$ the seeding rig from which the HFSBs are released into the flow coming from, $\boxed{8}$, the Open Jet Facility. (b) Schematic representation of the setup. The camera setup is mounted on a multi-axis linear actuator to allow movement along the $x$-axis.

The size of these smaller volumes determines the resolution of the reconstructed flow. Larger volumes, generally speaking, produce more consistent data at the cost of certain wake details, such as the tip vortices, which get absorbed into one large outer vortex in the averaging process. We used two cell volumes: $40 \times 40 \times 40$ mm$^3$ to analyse tip vortex behaviour, and $60 \times 60 \times 60$ mm$^3$ to calculate more general wake properties such as wind speed and energy advection. A 75 % overlap between volumes was chosen to have smooth transitions between subsequent volumes, resulting in a grid spacing of 10 mm and 15 mm, respectively.





Time-averaged velocity fields were acquired by binning the particles from all time steps following the previously described averaging process. To obtain time-varying flow fields, the particles from each time step can be binned separately. However, insufficient particles in parts of the volume can result in gaps in the flow fields. By averaging the particles for specific phases based on turbine measurements, such as the rotor azimuth position $\psi$, the number of particles used in the binning process increases, and the measurement uncertainty is reduced.

In the case of baseline operation, the phase averaging procedure consisted of binning the particles based on the rotor azimuth position into 12 bins of 30 degrees. Here, we assume that the wake dynamics are sufficiently represented by 12 discrete phase bins. Subsequent averaging for each of these phase bins resulted in consecutive flow fields that show the wake over a single rotor rotation. The Helix method complicates the phase-averaging procedure as the time-varying pitch actuation introduces additional dynamics to the wake that cannot be adequately captured in a single turbine rotation. Since the Helix actuation can be represented by a thrust force vector moving around the rotor plane, we introduced the Helix azimuth $\psi_h$ as an additional phase variable for the binning process (van der Hoek et al., 2024). More specifically, the actuation frequency of $St = 0.27$ was selected such that each Helix (and yaw) cycle coincides with 6 rotor rotations, i.e., $f_e/f_r = 6$. Hence, the wake dynamics of the Helix cases are represented by $6 \times 12 = 72$ phase-averaged flow fields.

## 2.4   Investigated Cases

In total, 6 different cases were investigated, of which 4 have different phase offsets, spaced $90°$ apart. All cases are summarised in Table 1. For the floating turbine that serves as the basis of this work, the phase differences can shift by $180°$ within the frequency range in which the Helix method is effective. The effect of this is investigated by including the $0°$, $90°$ and $180°$ phase-offset cases. The final case with a $270°$ is added to complete the measurement and provide more insight into the interaction between dynamic yaw and the Helix method. To best represent the motions a full scale turbine would undergo, the ratio between blade pitch amplitude and yaw amplitude are based on the identified input-ouput relation shown in Fig. 2. The measurement domain spans a distance of 4 rotor diameters, from $1D$ to $5D$ behind the turbine in steps of $0.5D$. Each measurement spans 400 mm in the $x$-direction and 800 mm in both the $y$- and $z$-directions. Since the width of a single measurement volume is larger than $0.5D$ there exists a small overlap between every measurement which aids with post-processing. Based on these measurements, the full three-dimensional wake can be reconstructed, enabling analysis of the interaction between the yaw motion of the floating turbine and the Helix wake mixing method. Using the PIV data, wake recovery, as quantified by the wake velocity, can be analysed. Furthermore, the same PIV data can also be used to analyse the behaviour of the wake, providing insight into the aerodynamic processes that occur behind the actuated turbine. Note that for these experiments, a single actuation frequency was chosen to limit the number of individual measurements, as a single wake measurement consists of multiple PIV measurements.



**Table 1.** Overview of measurement scenarios. For the cases with platform yaw motion, $\Delta\phi$ denotes the phase offset.

| Case name | $St$ | Blade pitch amplitude | Yaw amplitude | Phase offset |
|---|---|---|---|---|
| Baseline (no Helix) | 0.00 | $0.0°$ | Not applicable | Not applicable |
| Helix Bottom-Fixed | 0.27 | $\pm2.0°$ | Not applicable | Not applicable |
| Helix $\Delta\phi = 0°$ | 0.27 | $\pm2.0°$ | $\pm5.0°$ | $0°$ |
| Helix $\Delta\phi = 90°$ | 0.27 | $\pm2.0°$ | $\pm5.0°$ | $+90°$ |
| Helix $\Delta\phi = 180°$ | 0.27 | $\pm2.0°$ | $\pm5.0°$ | $+180°$ |
| Helix $\Delta\phi = 270°$ | 0.27 | $\pm2.0°$ | $\pm5.0°$ | $+270°$ |

## 3 Results

In this section, the results from the experiments are shown. First, the wind speed behind the turbine is analysed after which the energy entrainment is discussed. This is followed up by a detailed analysis of prominent wake structures and how these are
affected by the change in phase offset of the yaw motion.

An example of the results obtained during the measurement campaign is shown in Fig. 4. The wind speed is shown as velocity slices in the $x-z$ (left column) and $x-y$ planes (right column). Prominent vortex structures in the wake, typically tip and hub vortices, are visualised using three-dimensional iso-surfaces of the Q-criterion (Hunt et al., 1988; Soto-Valle et al., 2022). The threshold for isosurfaces is chosen such that the relevant wake dynamics are best visualised. Comparing the baseline
case with any of the Helix cases reveals distinct differences in wake dynamics. Where the baseline wake remains stable up to a distance of $4-4.5D$ from the turbine, the tip vortex structures are severely disturbed when the Helix is enabled. Furthermore, the wake is also dynamically deflected due to the Helix. Comparing the five Helix cases, we find that when the Helix input and yaw motion are in phase ($\Delta\phi = 0$), the wake deflection is enhanced. In contrast, when they are $180°$ out-of-phase, the deflection is reduced. In general, the structure of the tip vortices is noticeably different when the platform is yawing. If this
difference in wake structure has an impact on the effectiveness of a wake-mixing technique can be quantified by measuring the wind speed directly behind the turbine.

Figure 5 shows the time-averaged wind speed, normalised by the inflow velocity, that a hypothetical second turbine would experience when it is operating downstream of the first turbine. This is achieved by spatially averaging the time-averaged wind speed over the same area as the rotor disk. All cases show that the wake is recovering as it propagates downstream as indicated
by the increasing wind speeds. The increased mixing induced by the Helix method leads to an increased wind speed at the end of the domain compared to the baseline case. This gain of 6.6 percentage points (p.p.) in wind speed can be equated to an increase of 21 p.p. in the power available in the flow that a downstream turbine can potentially extract. Furthermore, when the platform is yawing in phase with the Helix input, an additional gain of 3.6 p.p. in wind speed is achieved, which translates into an increased power gain of 12 p.p. in the flow. When the yaw motion is $180°$ out-of-phase, the gain in wind speed is reduced
by 2.4 p.p., equating to a loss of 7 p.p. in terms of extractable power in the wake compared to the wake excited by the Helix method without any yaw motion.





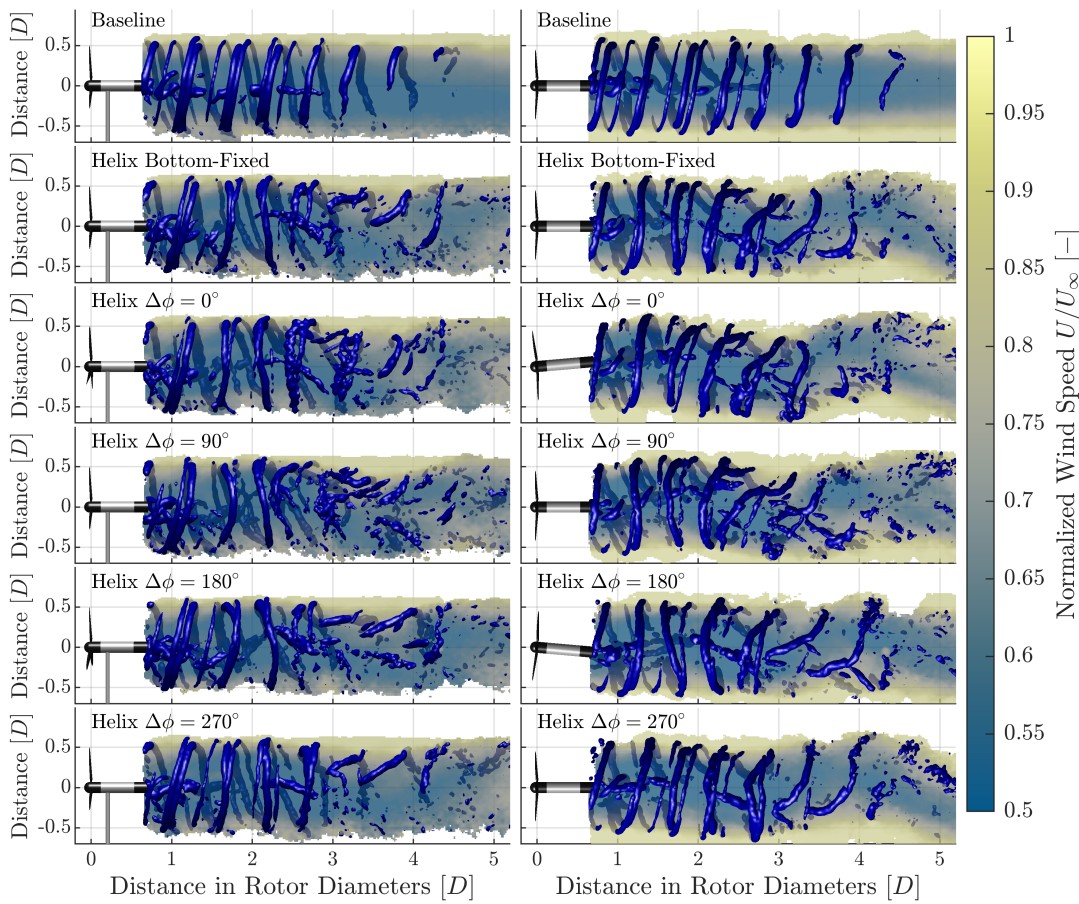

**Figure 4.** Reconstructed side view (left column: $x-z$ plane, $y=0$) and top view (right column: $x-y$ plane, $z=0$) streamwise wind speed slices and Q-criterion, represented by the blue iso-surfaces. For all cases, the data is taken halfway through a phase-averaged cycle.

An increase in wind speed equates to an increase in the kinetic energy of the wake. This energy is entrained from outside the wake boundary. This flux of kinetic energy is dominated by the Reynolds shear stresses (Reynolds and Hussain, 1972; Cal et al., 2010). The energy advection calculation is carried out in the radial direction over a control volume. This volume, whose boundaries are defined by the rotor surface, is schematically depicted in Fig. 6. Since the hexapod leaves a wind shadow below the wake, the bottom part of the wake is not considered for this analysis, as it is not representative for a full scale floating wind turbine.

Figure 7 shows the energy advection for the six different cases that are investigated. Analysis of the energy advection shows the same results as for the wind speed findings, i.e., when the platform is yawing in phase with the Helix input, energy advection is increased compared to the Helix case. When the yaw motion is $180°$ out-of-phase, the opposite holds. Furthermore, after a distance of three rotor diameters, energy advection becomes constant, and the differences between the individual Helix cases become smaller. From Fig. 4, it can be seen that this is, on average, also the distance where the tip vortex structures start to




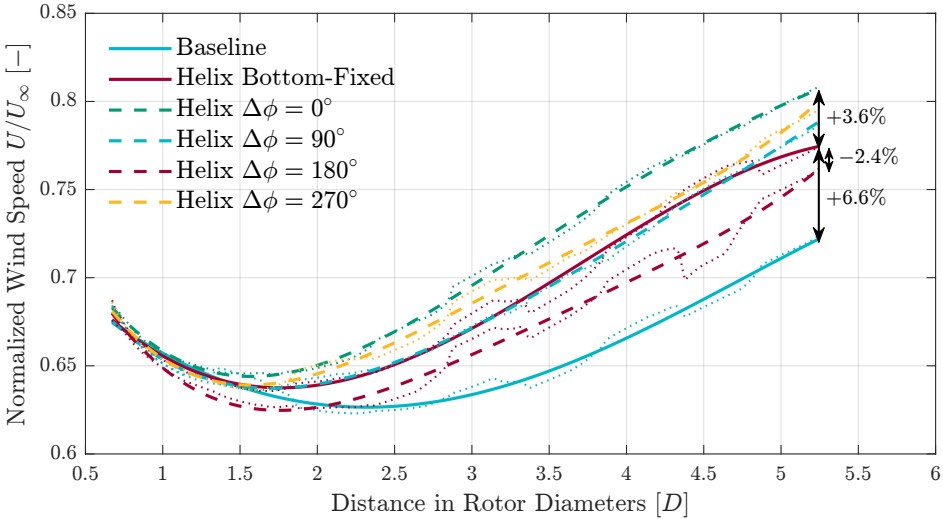

**Figure 5.** Rotor-average wind speed as perceived by a hypothetical downstream turbine in the wake. The thin dotted lines show the results as measured per individual measurement domain. This data is approximated using fourth-order polynomials (thick lines), removing the jumps in data between individual measurements. The solid lines represent the results without yaw motion, and the dashed lines represent the results when the turbine is undergoing yaw motion.

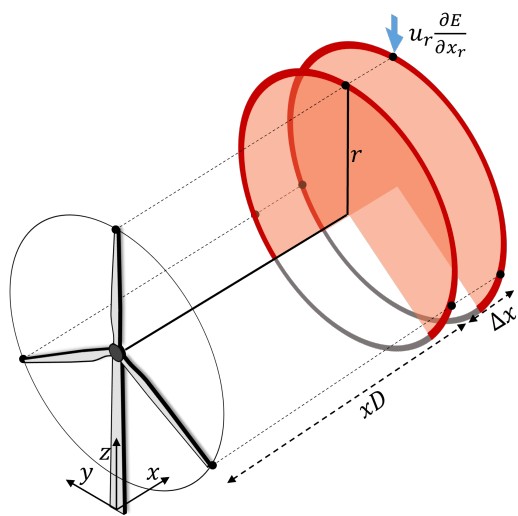

**Figure 6.** Schematics of the energy advection calculation in the radial direction over a control volume. We consider a ring (bright red) of radius $r$ located a distance of $xD$ from the turbine, over which the energy advection is calculated per downstream distance $\Delta x$. The Cartesian velocity components are first transformed to a cylindrical coordinate frame.



dissolve. Hence, the gain in wind speed, due to increased energy advection happens mainly in the area where the wake still is shielded from the ambient flow by the tip vortices and the mixing process has not fully started. The cumulative results, the total

energy entrained into the wake up to that point, support the finding that the in-phase case gains the most energy in the initial part of the wake. Any gains or losses in wake mixing dynamics therefore also happen close to the turbine.

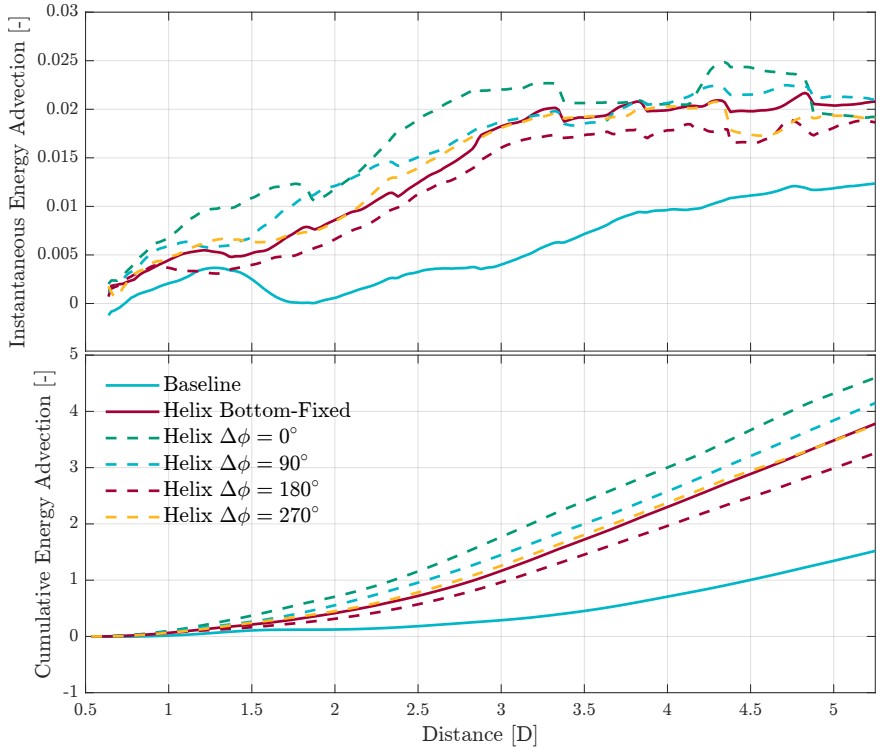

**Figure 7.** Local (top) and cumulative (bottom) results of the energy advection analysis using the phase-averaged data.

Studying the behaviour of the hub and tip vortices provides insight into the differences between the Helix method and the cases where the platform is yawing in and $180°$ out-of-phase. Figure 8a shows the tip vortices, visualised using iso-surfaces of the Q-criterion, and the location of the hub vortex indicated by red circles. The latter is traced using a Gaussian convolution

method (Coudou et al., 2018; Coudou, 2021). The left column shows the Helix at 4 different time instances $T$ within one cycle $T_p$ of the Helix excitation. The hub vortex starts to diverge from the centre at a distance of $2D$, interacting with the tip vortices at $3D$. Compared to the in-phase case (middle column), this behaviour is amplified when the turbine is yawing. The wake displacement is increased without altering the tip vortex structure until it starts to interact with the hub vortex. When the yawing is out-of-phase, the tip vortices are significantly more deformed, and the curvature introduced by the Helix is reduced.

Figure 8b shows the average radial distance for the tip and hub vortices with respect to the nacelle. As this value is calculated for exactly one cycle of the Helix and then averaged, the displacement of the wake as a whole is filtered out of the measurement. As such, the differences in the radial distance as shown in Fig. 8b stem from a difference in the interaction between the Helix




(a)

(b)

**Figure 8.** (a) Instantaneous tip vortices and hub vortex location for the Helix case (left column), Helix case with in-phase yaw motion (middle column) and Helix case with $180°$ out-of-phase yaw motion (right column). (b) Streamwise evolution of the averaged radial position for both tip and hub vortices with respect to the centre of the wake. The grey shaded area indicates the area in which wake breakdown is observed in the PIV data.





method and the dynamic yaw motion. Especially for the Helix case with in-phase yaw motion, the tip and hub vortex approach each other the fastest, followed by the Helix method and then the $180°$ out-of-phase yaw case. Moreover, when the hub and tip vortex are at the same radial distance they start to interact. This accelerated encroachment of the tip and hub vortex can be an explanation of the enhanced (reduced) energy advection shown in Fig. 7 when the platform is yawing in phase ($180°$ out-of-phase) with the Helix method. Further research into this behaviour using, for example, large eddy simulation methods could shed more light on the nuanced differences that happen with these interactions. What is clear from these results is that specific floating turbine dynamics can have a significant impact on aerodynamic processes that happen further in the wake, and that they are coupled.

## 4 Conclusions

This work demonstrates how the dynamics of a floating turbine interact with that of the Helix wake mixing method. The presence of a natural frequency in the yaw motion for certain types of foundations can lead to different phase couplings between control input and floating turbine dynamics. By experimentally analysing the 3D wakes and aerodynamics of a floating turbine model, we find that actuating the Helix at a frequency such that the yaw motion is in-phase results in a significantly better wake recovery than when the turbine yaws $180°$ out-of-phase. Analysing the energy advection into the wake indicates that for the in-phase case, significantly more energy is transferred into the wake between a distance of 1 to 3 rotor diameters downstream. A significant reduction is found for the $180°$ out-of-phase case.

Using the volumetric PIV measurements allows us to visualise the location of the tip and hub vortices, revealing that the dynamic interaction between the two is influenced by the platform yaw motion. The earlier interaction between the tip and hub vortex leads to an earlier breakdown of the wake, accelerating the energy advection into the wake. When yawing at $180°$ out-of-phase, this interaction is both reduced and delayed, explaining the reduced effectiveness of the wake mixing method. Further investigations should include analysing different phase offsets, as it could well be that the optimal offset lies between the investigated values.

This work shows that the dynamics of a floating turbine can be effectively used to enhance the performance of wake mixing controllers. These outcomes can be used to design floating turbines that optimise both control and turbine design, a process called control co-design. These optimal designs could also be extended to tackle different control challenges for floating wind turbines. For example, the pitch instability of a floating turbine manifests itself in different weather conditions to those when the Helix is effective. A single, optimised controller could account for both control challenges. This will significantly contribute to the development and deployment of advanced 'smart' floating wind farms.

## Appendix A: Further Investigation into Floating Turbine Dynamics

This supplementary material provides background information on the frequency identification techniques used to determine the floating turbine dynamics. This analysis is carried out in QBlade (Marten, 2022), a simulation suite capable of simulating





hydro-, aero-elastics and wake dynamics. To capture the dynamics of the OC4 (Robertson et al., 2014), TripleSpar (Lemmer
et al., 2018) and VolturnUS-S (Allen et al., 2020), three different semi-submersible floater foundations each of them is excited
using a chirp signal on the blade pitch angle. Specifically, since we are interested in how the Helix excites the yaw dynamics
we apply the chirp signal to the $\beta_{yaw}$ fixed frame pitch angle. The fixed frame pitch angle is actuated with an input frequency
of $1 \cdot 10^{-3}$ to 1 Hz, which covers the frequency range in which the Helix is typically applied. The input is logarithmically
distributed over the full duration of the experiment such that more of the data collected is generated at lower frequencies. The
duration of the experiment is 28800 seconds (or 8 hours), sampled at 0.05 seconds. The long duration ensures enough data is
present to identify low-frequency dynamics accurately. Based on the input-output relations it is possible to deduce how much
yaw movement a sinusoidal time-varying blade pitch input creates. An example of such an input-output relation is given in Fig.
A1, which shows the blade pitch angle input and the yaw motion of the TripleSpar as output.

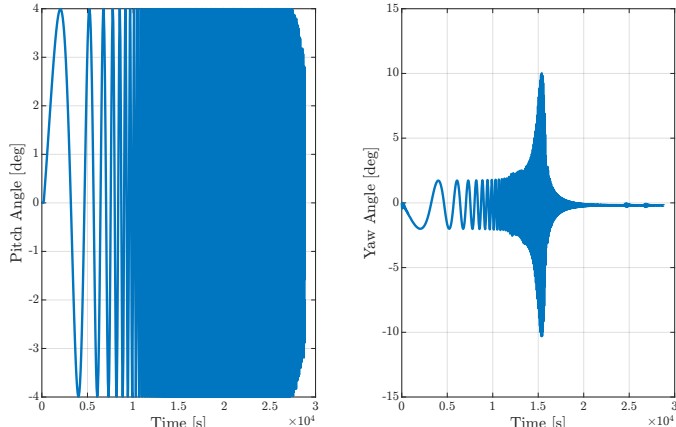

**Figure A1.** An example of the identification input used for system analysis. On the left is the pitch angle input and on the right is the yaw
angle of, in this particular case, the TripleSpar platform. The large increase in yaw motion coincides with the eigenfrequency of the floating
turbine.

**Frequency-Domain Analysis**

The input-output data as seen in Fig. A1 can be used to derive the frequency response functions (FRFs) in terms of gain as
well as phase relations. These FRFs are identified using system identification tools developed in (Van der Veen et al., 2013).
Figure A2 shows these response functions for the OC4, TripleSpar and VolturnUS-S platforms. The shaded areas in Fig. A2,
with the same colour as the graphs, indicate the frequency areas for $S_t \in [0.2\ 0.4]$, the frequency range in which the Helix
method is most effective (van der Hoek et al., 2024; Mühle et al., 2024). The three different foundations each mount differently
sized turbines, ranging from $D = 126$ m for the OC4 mounting the NREL 5MW turbine (Jonkman et al., 2009), $D = 178.3$ m
for the DTU 10MW turbine (Bak et al., 2013) on the TripleSpar platform, to $D = 240$ m for the IEA 15MW turbine (Gaertner



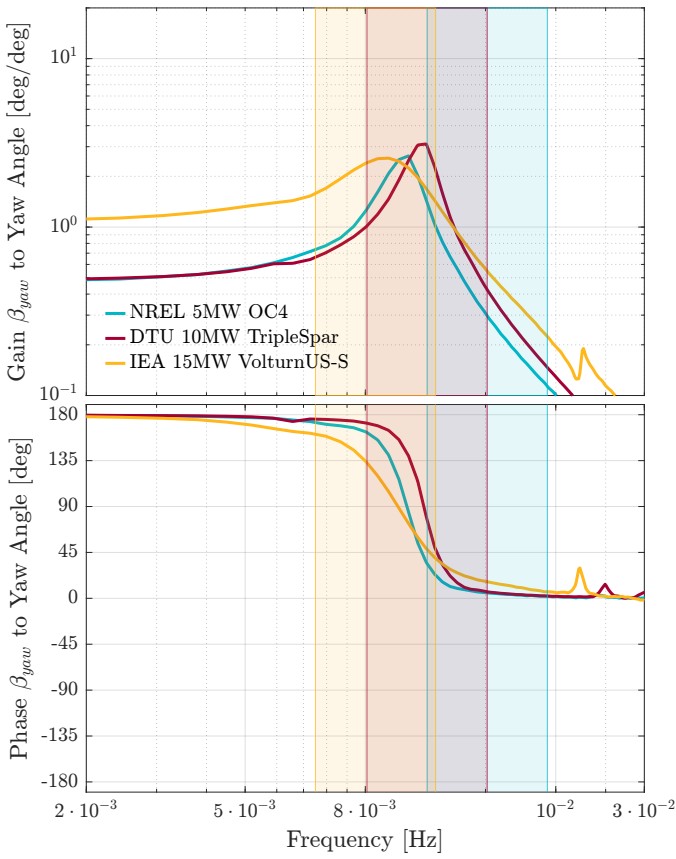

**Figure A2.** Frequency response functions for yaw motion when excited by a fixed-frame yaw pitch angle. The shaded areas denote the frequency range in which the Helix method provides significant wake mixing. The colour of the shaded area is matched to the bode plot colour.

et al., 2020a) mounted on the VolturnUS-S floater. All three turbines are designed to operate in similar conditions and for similar water depths, hence the similar dynamics, even though the turbines vary significantly in size.

Each floating turbine has a similar gain in yaw motion, varying from a peak gain of $2.5°$ per degree of blade pitch input
for the VolturnUS-S foundation to $4°$ for the TripleSpar foundation. Coincidentally, the eigenfrequencies coincide with the actuation frequencies of the Helix method. Especially for the TripleSpar foundation the eigenfrequency is exactly in the Helix frequency range. The work in the paper focuses on the impact of phase offset on the effectiveness of the Helix wake mixing method. Within the operational range of the Helix, three different phase couplings can occur. Furthermore, the frequency of the Helix can also change as the Strouhal number is dependent on the wind speed, see Eq. (1). A change in wind speed without
changing the input frequency will result in a different actuation frequency, and thus potentially different dynamics.

This analysis is carried out with no sea state, i.e., there are no waves present. For the wind conditions at which the Helix is beneficial, waves will be present (Hasselmann et al., 1973). The interaction with these waves will also instigate motions in



the floating turbine. However, at below-rated wind conditions for which the Helix is beneficial, these motions are significantly
smaller than those excited by the Helix method itself. Furthermore, the frequency of wave excitation is typically an order of
magnitude higher than the application frequency of the Helix. For floating structures response amplitude operators (RAOs)
describe how the vessel responds to wave inputs (Journée et al., 2015). The RAOs for the IEA 15MW VolturnUS-S floating
turbine can be found in (Allen et al., 2020). Finally, for this work, the assumption is that the wind is always perfectly aligned
with the turbine. Waves are typically a result of the wind sweeping over the water surface. As such the waves will also be
aligned with the floating turbine (Young, 1999), resulting in no excitement in yaw motion.

*Code and data availability.* The data used in this work are available at the 4TU repository:
https://doi.org/10.4121/a8555119-db46-4ecd-9138-8785b9080ff0.v1.

*Author contributions.* D.vd.B and D.vd.H conceived the wind tunnel experiments presented in this work. They also performed the experiments and post-processed the data. The paper was written with J.G., D.D.T and J.W.v.W, and its content was extensively discussed between all authors.

*Competing interests.* The authors have the following competing interests: At least one of the (co-)authors is a member of the editorial board
of Wind Energy Science.

*Acknowledgements.* This project is part of the FLOATECH project and its follow-up project named FLOATFARM. The research presented in this paper has received funding from the European Union's Horizon 2020 research and innovation programme under grant agreement No. 101007142 and the European Union's Horizon programme under grant agreement No. 101136091.





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
