# Peer review of "Phase-controlling the motion of floating wind turbines to reduce wake interactions"

_Wind Energy Science, 2025_

## Author Comment (AC1)

| Date | December 18, 2025 |
|---|---|
| Contact person | Daan van der Hoek |
| E-mail | D.C.vanderHoek@tudelft.nl |
| Subject | Response to reviewers |

**Delft University of Technology**

Delft Center for Systems and Control

Address
Mekelweg 2 (ME building)
2628 CD Delft
The Netherlands

www.dcsc.tudelft.nl

Reviewer #1, Reviewer #2
*Wind Energy Science*

Dear Reviewers,

We thank you for all the constructive comments that we received on the paper. We are confident that these comments have helped us to improve the quality of the paper significantly. In response to your feedback, we have made several revisions to the original manuscript. In this document, we aim to address all points raised by the reviewers. All changes made to the paper are indicated in italic font. An additional document has been attached to our response, highlighting all the changes made to the paper.

Yours sincerely,

Daniel van den Berg,
Daan van der Hoek
Delphine De Tavernier
Jonas Gutknecht
Jan-Willem van Wingerden

Enclosure(s):  Response to comments of Reviewer #1
Response to comments of Reviewer #2

**Response to comments of Reviewer #1**

General comments: The manuscript presents wind tunnel experiments on a dynamically yawing scaled wind turbine platform, designed to emulate the motion of a floating offshore turbine subjected to helix-type control inputs. The authors show that wake recovery is enhanced when the helix actuation is in phase with the yaw motion, and they provide an explanation based on vortex interaction mechanisms.

Overall, the paper is clearly written, the figures are of high quality, and the topic is timely and relevant to ongoing research in floating wind turbines and wake control. The results are novel and potentially valuable. However, before publication, several aspects—particularly regarding the physical modeling, scaling assumptions, and the description of the QBlade simulations—need clarification or expansion.

Below, I provide major points of discussion followed by specific comments where I refer to the lines of text in the preprint pdf.

I hope they can be useful to improve this work, and I look forward to see a new iteration.

Major Comments:

1. *Description of the QBlade Simulations and Motion Scaling*
   The paper's methodology relies heavily on simulated platform motions obtained with QBlade, which are then prescribed to the moving platform in the wind tunnel. Currently, this description is in the Appendix and lacks clarity. I strongly suggest moving this material into the main text and expanding it. In particular: the simulations use multi-megawatt rotors and realistic floating platforms. How are these motions scaled to the MoWiTo model? Please discuss the scaling laws (Reynolds, Froude) and justify whether dynamic similarity is preserved.

   We agree with the reviewer that the original manuscript lacks clarity in some parts. In particular, it is not clear whether the full-scale turbine simulations are part of the current study. In fact, the simulation results in Fig. 2 and the description in the Appendix are from previous studies by the authors. To clarify this, we have decided to remove the Appendix from the manuscript and refer to the respective papers. We believe this will improve the clarity of the paper.

   The results from one of these studies, presented in Fig. 2, show how different platform motions are excited by the application of the Helix method. Based on the figure, we conclude that the yaw degree of freedom (DOF) is most affected by this, and that the pitch and tilt DOFs are barely excited. Although we do not show it here, the same goes for the other platform DOFs. Note that these simulations also considered realistic waves. Therefore, we simplified the experimental setup by only considering the yaw motion of the turbine.

   Since the yaw motion is excited at the same frequency as the Helix method, the only scaling law we consider is the Strouhal number. Wind tunnel studies with small-scale turbines generally suffer from lower Reynolds numbers, especially considering the blade chord length. The blades of the MoWiTO were designed specifically for low Reynolds numbers. The large-scale shape of the wake, however, becomes independent of Reynolds number from $Re > 9.3 \times 10^4$ [2], which is the case for the MoWiTO in this study ($Re \approx 1.9 \times 10^5$).

   Section 2.2: *"These settings result in a rotor diameter based Reynolds number of $Re_D = U_\infty D/\nu \approx 1.9 \times 10^5$, which ensures wake similarity to full scale turbines (Chamorro et al., 2012)."*

   It appears that only yaw motion is modeled, while wave-induced and tilt motions are neglected. Please clarify whether this simplification is justified, and discuss its implications for the generality of the results.

   The reviewer is correct. Simulation results from previous studies have shown that for some floating platforms, such as the VolturnUS, only the yaw motion was excited by the Helix method (see Fig. 2 of the paper). Since the primary purpose of this paper is to validate whether there is a coupling between the Helix method and dynamic platform yaw, we have chosen not to model the other platform motions. More details are provided further on in response to specific questions.

2. *Impact of the low tip speed ratio*
   The scaled turbine operates at an optimal TSR of 5, which is substantially lower than that of modern large turbines (typically 7–9). The paper should discuss how this affects the wake development and vortex dynamics.
   In Fig. 7, the near-wake differences between in-phase and anti-phase cases are significant; however, with a higher TSR the near wake would likely contract and recover differently.
   Similarly, the characterization of tip and root vortices (Fig. 8) should note that vortex strength and persistence depend on TSR.
   We agree with the reviewer that a higher tip-speed ratio would impact the results presented in this paper. We have added some comments in the results section to reflect on this. Please see our more detailed response to some of your other related comments below.

3. *Possible Complementary CFD Analysis*
   The results could be strengthened by comparison with or validation against CFD simulations (e.g., LES ALM). Even a qualitative comparison would enhance the credibility of the wake structure analysis.
   The main motivation for this paper was to obtain experimental proof for observations that were made in previous simulation studies using a free vortex wake method [7, 5]. While we agree that additional simulations with LES could further strengthen these previous studies and provide new insights, we believe this falls outside the scope of the current paper and will be reserved for future work. This recommendation is now mentioned in the conclusion.
   *"High-fidelity large eddy simulations can also investigate whether the difference in phase offset will play a significant role with higher ambient turbulence"*

More specific comments:

- Title: From the title, it is not clear that the helix is being applied or that dynamic yaw is being investigated.
  We agree with the comment and have decided to change the title to make this clear:
  *"Phase-controlling the yaw motion of floating wind turbines with the Helix method to reduce wake interactions: an experimental investigation"*.

- Introduction: a discussion on the state-of-the-art research for dynamic yaw is missing. In fact, dynamic yaw has also been studied as a mixing technique. As such, it features its optimal actuation frequency and amplitude. It would be interesting to include an assessment of this with regard to your experiments.
  We have added the following sentences concerning dynamic yaw control to the introduction.
  *"Applying the Helix method on such a floating turbine results in the platform starting to yaw dynamically. Dynamic yaw control is also a wake mixing control method, and when applied with the right frequency and amplitude, it can improve wind farm power yield (Lin and Porté-Agel, 2024; Mühle et al., 2024a). A combination of these two methods (i.e., the Helix method and dynamic yaw) could potentially further enhance wake recovery."*

- Line 18: Why can a wind farm experience an efficiency drop? I guess it is because of the wake effects. Please clarify in the text.
  You are correct, we have adjusted the sentence accordingly (in bold):
  *"Although individual turbines are capable of operating close to their theoretical maximum efficiency, a wind farm can experience an efficiency drop of up to 40 %* **due to the interaction between wind turbines** *(Bastankhah and Porté-Agel, 2016; Howland et al., 2019; Barthelmie et al., 2010)."*

- Line 19: sentence could be a bit more formal/technical.
  Sentence has been rewritten:
  *"As a wind turbine extracts energy from the incoming airflow, it leaves a wake of lower velocity and more turbulent airflow."*

- Line 23: There are articles around where wake steering through floating platform motion was investigated, but from this sentence, it seems like something like this has never been done.
  Thank you for pointing this out. We acknowledge the research that has been done on wind farm control for floating applications. This sentence is intended to convey that most research focuses on bottom-fixed applications. We have now added a reference to wind farm control for floating wind farms.
  *" With some exceptions (e.g., Kheirabadi and Nagamune, 2020), the development of control approaches to mitigate wake interactions has so far focused mostly on bottom-fixed wind farms (Meyers et al., 2022; Houck, 2022)."*

- Figure 1: Please add in the caption what is being shown (i.e., Q-criterion), and how this was obtained (i.e., LES ALM, CFD, ...).
  Additional information has been added to the figure caption (in bold):
  *"Model of the IEA 15MW turbine mounted on the VolturnUS-S floater, with the left figure showing the wake when using a baseline controller, and on the right figure the distorted wake when the Helix method is enabled.* **The visualised wakes feature iso-surfaces of the streamwise velocity taken from large eddy simulation results by Frederik et al. (2020b)."**

- Line 45: $f_e$ is introduced as the actuation frequency. With the Helix method, however, there is an ambiguity because the actuation frequency of the pitch motors is different from the rotation frequency of the thrust vector. Please clarify what $f_e$ is.

  Thank you for addressing this. In this case, we are referring to the rotation frequency of the thrust vector. We have added the following sentences to clarify this:
  "With the Helix method, $f_e$ refers to the frequency at which the thrust vector circles the rotor plane. To achieve this, the blades have to pitch at a much faster rate, more specifically $f_\beta = f_r + f_e$, with blade pitch frequency $f_\beta$ and rotor frequency $f_r$."

- Figure 2: What turbine was used in these simulations? What ambient conditions? Is this a scaled result? Perhaps the x-axis could be normalized?

  We now realize the original manuscript did not make it clear enough that this figure is based on simulations from a previous publication. It is based on simulation data of a full-scale turbine [6]. The text and figure caption now contain some additional information and refer to the related paper.
  "The results presented in Fig. 2 were obtained through identification experiments by van den Berg et al. (2024b) using a full-scale floating turbine (IEA 15MW on the VolturnUS platform). For this experiment, the inflow was set to be uniform and constant at 9 ms$^{-1}$. The wave conditions were chosen such that they correspond to calm weather at that wind speed."

- Line 52: As I mentioned earlier, the dynamics of the floating platform must be well modeled, since it is the foundation of your work. I would move the appendix section to a dedicated, expanded section where Figure 2 could be inserted.

  We agree with the reviewer's comment on the importance of realistic modelling. The results presented in this figure were obtained through aero-servo-hydro-elastic simulations with QBlade [4] using realistic wave conditions for a constant uniform wind speed. The purpose of Fig. 2 was to highlight the effect of the helix method on a platform motion and phase that was observed in previous studies. For that reason, we have decided to remove the appendix and refer to the respective papers instead.

- Line 59: The energy amount needed to actuate the blades with the helix is important. Can you quantify how much energy is "some energy", as a fraction of rated power, for instance?

  This sentence referred to the efficiency losses that are introduced by the continuous pitch motion of the blades. Different LES studies have shown that these losses are generally in the order of 1 to 3%, depending on the pitch amplitude. The text has been adjusted accordingly.
  "Although the application of the Helix methods and the resulting movement introduce small efficiency losses to the upstream turbine (i.e., power losses associated with the Helix method are generally in the range of 1–3% (Taschner et al., 2023)), ..."

- Line 60: can you provide a number for how much the power generation is increased?
  We have now included a number.
  *"..., van den Berg et al. (2024) found that the power generation of a two-turbine wind farm can increase up to 8% when the floating turbine yaws with the Helix method applied."*

- Line 67: "The wind tunnel experiments are carried out that combine a hardware...": this sentence is not clear.
  The sentence has been changed for clarity:
  *"The wind tunnel experiments combine a hardware-in-the-loop setup with tomographic particle image velocimetry (PIV) to measure the effect of the Helix method and yaw motion on the wake."*

- Line 70: the yaw motion is prescribed, but what about the effects of waves, other DOF other than yaw, and other higher frequency motion? Please discuss why they were omitted, and the limitations associated.
  This is a good point that we did not mention in the original manuscript. The other DOFs were omitted for two reasons: 1) This would have required us to rescale the effect of wave excitation on the hexapod motion, which uses a different scaling factor than the yaw motion induced by the Helix method, which is based on the Strouhal number. Therefore, it would add considerable complexity to the experimental setup. 2) By only considering the yaw motion, we can isolate its phase impact on wake recovery without having to worry about the influence of other motions. In reality, these motions will also affect the wake recovery, and therefore the potential gain with the Helix method. But they do not affect the coupling of the Helix method and the yaw motion.
  Section 2: *"The yaw motion is imposed on the hexapod, with the motion being representative of an actual floating turbine applying the Helix method according to the dynamics shown in Fig. 2. The pitch and roll motions of the platform are not considered, as Fig. 2 shows that these motions are not strongly affected by the Helix method. The platform motion due to waves is not reproduced for simplicity and to isolate the effect of the Helix and dynamic yaw motion on wake recovery."*

- Line 94: Please report the blockage ratio: Is blockage biasing your experiments in any way? Especially if the rotor is placed in proximity to the jet outlet, I expect that there could be some blockage.
  We have added the following sentence to Section 2.2.:
  *"The blockage ratio based on the rotor swept area and jet outlet is 3.3%, requiring no corrections to the wake measurements (Chen and Liou, 2011).*

- Line 96: Are 2 kHz enough to capture the floating platform dynamics, considering time scaling?
  Since we only consider the yaw motion of the turbine associated with the Helix excitation frequency, which is in the order of 2Hz, the current sampling frequency is sufficient.

- Line 107: discuss the implications of a rather low tip speed ratio.
  This is a valid point. A higher tip-speed ratio results in faster interaction between tip-vortices, leading to earlier wake recovery [3]. One can therefore expect that with a higher tip-speed ratio, the entrainment in Fig. 7 starts earlier in all cases. However, the relative increases in wake recovery with respect to baseline may differ. We do not expect the coupling mechanism between the Helix method and dynamic yaw to change as a result of this, as this phenomenon was previously observed in simulations with Qblade. However, additional simulations with LES could strengthen this study. We have added the following sentences to the discussion of the results.
  *"It should be noted that the turbine used in this study has a lower tip-speed ratio than full-scale turbines. Given that higher tip-speed ratios generally result in earlier wake recovery (Lignarolo et al., 2015), the results presented in the previous figures may differ for full-scale turbines. However, we do not expect to see any changes in the coupling behavior of the Helix method with dynamic yaw. Further research into this behavior, using methods such as large eddy simulations, could shed more light on the nuanced differences that occur with these interactions."*

- Section 2.4: How was the helix amplitude chosen?
  The helix amplitude in the experiment is mainly limited by the bandwidth of the pitch actuators. At the current amplitude and frequency, the Helix can achieve a power gain while not overexerting the actuators. Similar settings were used in the experiments of [8].
  Section 2.4: *"The blade pitch amplitude for the Helix was set to 2°, similar to (van der Hoek et al., 2024)."*

- Section 2.4: The phase offset between yaw and helix is not defined very clearly to me. Perhaps you could add some equations that clarify what $\Delta Phi$ is.
  We added the following definition to Section 2.4.
  *"The phase offset $\Delta\phi$ is defined as the phase difference between the yaw moment from the Helix $M_{yaw}$ and the yaw angle $\gamma$."*

- Line 166: could this be explained by a study of the instantaneous deflection of the thrust vector due to yaw and due to the helix?
  We believe this is correct, when the yaw motion and helix are in phase, the instantaneous deflection from the thrust is amplified, and vice versa for the out of phase case.

- Figure 5: It would be nice to see errorbars here.
  By adding error bars to all lines, the figures became a bit too cluttered to see clearly. Therefore, we added a subplot that shows the mean and standard deviation intervals of all cases measured at a distance of $5D$ downstream. In this case, the standard deviation denotes the rotor-average velocity fluctuations over a single Helix cycle. We hope that this addition has improved the figure.

- Figure 7 and 8 and relative analysis: I wonder how these results would change with a higher operational tip speed ratio.
  With a higher tip-speed ratio, the distance between consecutive vortices will be shorter, and their strength will be greater. This will cause the vortices to start interacting at an earlier stage, resulting in earlier wake recovery [3]. We expect the earlier wake recovery to be visible in Figs. 7 and 8. However, we do not expect to see any changes in the interaction with the Helix and dynamic yaw.
  *"It should be noted that the turbine used in this study has a lower tip-speed ratio than full-scale turbines. Given that higher tip-speed ratios generally result in earlier wake recovery (Lignarolo et al., 2015), the results presented in the previous figures may differ for full-scale turbines. However, we do not expect to see any changes in the coupling behavior of the Helix method with dynamic yaw. Further research into this behavior, using methods such as large eddy simulations, could shed more light on the nuanced differences that occur with these interactions."*

- Line 177: How is the instantaneous energy advection defined? Please add a formula.
  Throughout the paper, we used the term energy advection to indicate the amount of energy that is entrained from the free stream flow into the wake. A more precise definition would be entrainment or the mean flux of kinetic energy due to the turbulence [1]. This definition has been added to the paper in Equation (4), and the term energy advection has been replaced throughout the text.

- Line 180: The reason why the lower part of the wake is neglected is understood. However, there is no discussion on how big is the sector that is left out (aperture angle), and why it was made this way. This is not a good practice, since it seems an arbitrary choice. Perhaps by comparing with older measurements, the impact of the hexapod could actually be quantified and accounted for. Or LES simulations could be used to clarify this point.
  This should indeed be mentioned in the text. A cross-section of the wake is given in Fig. 1 of this document. This figure shows the streamwise velocity at a distance of $1D$ downstream. In the bottom quarter of the wake, we can see the outline of the tower. The hexapod itself is located further down and is not visible in the plot. However, as a precaution, we decided to include only the area indicated in the figure in the analysis of Fig. 7 in the paper. The entire lower quarter, between $\theta = -45°$ and $\theta = 225°$, of the wake has been left out of the analysis. We have revised the text to clarify this point.
  *"Since the hexapod leaves a wind shadow below the wake, the bottom part of the wake (between $\theta = 45°$ and $\theta = 225°$) is not considered for this analysis as a precaution."*

- Line 190: How general are these results if the effectiveness of the in-phase actuation is limited to the near wake?
  We agree that the phrasing in this sentence might give a wrong conclusion. The point we want to make is that the additional wake recovery of the in-phase case is achieved in the near wake region. After $3D$, the entrainment levels off and is closer to the other cases. However, due to this better start, the overall wake recovery (cumulative) remains ahead of the other cases for the entire length of the measured wake. We have adjusted the text to make this clearer.
  *"Hence, the gain in wind speed, due to increased entrainment, happens mainly in the area where the wake is still shielded from the ambient flow by the tip vortices, and the mixing process due to random fluctuations has not fully started. The cumulative results, the total energy entrained into the wake up to that point, support the finding that the in-phase case, compared to the other cases, gains the most energy in the initial part of the wake. This head start of the in-phase case allows it to stay ahead of the other cases for the entire measured wake."*

- Line 200: How was the tip vortex tracked in Figure 8b? The procedure for the root vortex is specified, but not the one for the tip vortex.
  Thank you for noticing this. The method for tracking the tip vortices is actually very similar. A Gaussian convolution is also used, but we restrict it to the outer ring of the wake. We adjusted the text as follows:
  *"The locations of both the tip vortices and hub vortex are tracked over time using a Gaussian convolution method (Coudou et al., 2018; Coudou, 2021)."*

- Line 205: "encroachment" is a bit difficult word. Consider replacing with a more common synonym.
  The sentence has been modified.
  *"This accelerated interaction between the tip and hub vortex ..."*

- line 232: "a simulation suite capable of simulating", remove repetition.
  The appendix and relevant text have been removed from the paper.

- line 234: what are OC4, TripleSpar, and VoltrunUS-S?
  The names refer to different floating platform designs, of which each will have a different interaction with the Helix. For example, the input-output relation shown in Fig. 2 is based on the VolturnUS platform, but other platforms may exhibit different dynamics.

- line 261: Why is the future tense used here and later?
  The appendix and relevant text have been removed from the paper.

- lines 264, 265: lack of citations for "... these motions are significantly smaller than those excited by the Helix method itself.", and "the frequency of wave excitation is typically an order of magnitude higher than the application frequency of the Helix". These are rather important statements that require a citation or a discussion, in my opinion.
  These statements were based on previous studies [7, 5]. However, we have decided to remove the appendices as mentioned previously.

- Side note: if the MoWiTo is equipped for it, it would be useful to quantify the fatigue and actuator duty cycles for the different scenarios investigated.
  The MoWiTO was equipped with strain gauges to monitor the fore-aft tower bending moment. Unfortunately, it was not possible to reconstruct these measurements for the dynamic yaw cases due to the strong yaw motions of the hexapod.

[Figure]

Figure 1: Streamwise velocity in the wake of the MoWiTO at a downstream distance of $1D$. The black lines indicate the part of the wake that is considered in the analysis of Fig. 7 in the paper.

**Response to comments of Reviewer #2**

The paper presents a wind tunnel experiment investigating how the yaw motion of the floater of a floating offshore wind turbine affects wake recovery when combined (or not combined) with HELIX wake control.

The paper is well written, and the study is highly relevant to the wind energy community, as it addresses the important topic of wind turbine wake control.

Therefore in the reviewer's opinion it deserves publication to WES after the following listed points are addressed.

1. An important consideration in wind-tunnel wake measurements is the influence of ambient turbulence. In the present experiment, the ambient turbulence level is relatively low (0.5%–2%), considerably lower than what is typically observed in the field. As a result, although the qualitative trends remain consistent, the quantitative conclusions may differ under real conditions, where higher ambient turbulence enhances and accelerates wake mixing. It is recommended to the authors to address this point in the introduction and perhaps in the conclusion section and discuss the limitations. Please indicate the lessons learned by the experiments you cite in the introduction section with regard to the above point.

   This is a valid comment that the original manuscript did not address sufficiently. Higher levels of atmospheric turbulence will impact the performance of the helix method with dynamic yaw, mostly due to accelerated recovery of the baseline wake. We have added a paragraph to Section 3 that discusses this point.

   *"The results of this study were obtained in low turbulence, with ambient turbulence intensity levels between 0.5% and 2.0%. Full-scale floating wind turbines will experience slightly higher levels of atmospheric turbulence. Therefore, we expect that the performance increase with the Helix method, and the coupling with dynamic yaw, will be lower in such a setting. This reduction is primarily due to the enhanced wake recovery in the baseline case that is associated with increased turbulence (Korb et al., 2023)."*

2. Figure 2, which presents the frequency-response analysis of the realistic dynamic system of a floating offshore turbine, indicates that the phase difference between the pitch motion and the platform's yaw motion is closer to 180 deg than to 0 deg. In contrast, the experimental results suggest that mixing is enhanced when the two motions are in phase. The frequency plots in Figure 2 represent the response of a dynamic system that cannot be subject (at least not easily) to modulation. Therefore, while it is interesting that in-phase motion appears to enhance mixing, it may be unrealistic to assume that such perfectly in-phase motion can occur in practice. Please comment on this issue.

   The reviewer is correct that not all phase offsets that were studied can be achieved by the floating platform presented in Fig. 2. However, within the effective Strouhal range shown in the figure, we can select the most optimal phase offset, and prevent the platform from yawing at a suboptimal phase. The platform in Fig. 2 serves merely as an example to illustrate how the coupling between the Helix method and platform yaw changes depending on the actuation frequency. Other platforms will have different dynamics and might have more beneficial phase relations. Furthermore, we can even incorporate these aspects in the design phase of the floating platform. We have added a paragraph at the end of the results section regarding this.

   *"Returning to the dynamics of the floating platform in Fig. 2, we notice that not all phase offsets studied in this experiment are present. However, we can choose the actuation frequency of the Helix method to obtain the most optimal phase offset for the floating platform, within the effective Strouhal range. Depending on the wind turbine model and platform type, the magnitude and phase relation of the platform motion and the Helix method can differ significantly. Some platforms may offer more advantages to the Helix method, whereas others may yield opposite results. By considering these effects in the floating platform design phase, we can even optimize the design for the Helix method (van den Berg et al., 2024a)."*

3. A detailed wake measurement campaign was carried out, and it is unfortunate that the paper presents only the averaged cross-sectional wake velocities. It would be helpful if the authors included a plot showing the spatial variation of velocity within the wake, illustrating how the enhanced recovery observed in certain cases is distributed across the wake region. Such data could serve as benchmark results for validating numerical models. The pattern plots of figure 4 do not provide that necessary quantitative detail.

   The flow fields presented in Fig. 4 show dynamic flow fields, including the spatial variation of the wake. This is particularly noticeable on the right-hand side of the figure for the Helix cases. For a more detailed analysis of the dynamic wakes, we refer to the online database mentioned in the paper, which features the three-dimensional phase-averaged wake measurements.

4. Please provide the formulas used to calculate the energy advection, along with details on how this quantity is computed.
   Please also refer to our response to reviewer 1 concerning a similar comment. An equation for the mean flux of kinetic energy due to turbulence (entrainment) has been added to the text (Eq. 4).

5. For the sake of completeness and clarity, please indicate whether the picture in Figure 1 is a schematic representation of the wake or a result of simulations and of what kind of simulations.
   Some additional information has been provided in the figure caption (in bold):
   *"Model of the IEA 15MW turbine mounted on the VolturnUS-S floater, with the left figure showing the wake when using a baseline controller, and on the right figure the distorted wake when the Helix method is enabled.* **The visualised wakes feature iso-surfaces of the streamwise velocity taken from large eddy simulation results by Frederik et al. (2020b)."**

6. Please rephase the following sentence which is unclear (in Section 2, second sentence): "The wind tunnel experiments are carried out that combine a hardware-in-the-loop setup and tomographic particle image velocimetry (PIV) to represent and measure the turbine dynamics, the coupled hydrodynamics, and the resulting aerodynamics."
   The sentence has been changed for clarity:
   *"The wind tunnel experiments combine a hardware-in-the-loop setup with tomographic particle image velocimetry (PIV) to measure the effect of the Helix method and yaw motion on the wake."*

**References**

[1] R. B. Cal, J. Lebrón, L. Castillo, H. S. Kang, and C. Meneveau. Experimental study of the horizontally averaged flow structure in a model wind-turbine array boundary layer. *Journal of Renewable and Sustainable Energy*, 2(1):013106, 1 2010.

[2] L. P. Chamorro, R. E. Arndt, and F. Sotiropoulos. Reynolds number dependence of turbulence statistics in the wake of wind turbines. *Wind Energy*, 15(5):733–742, 7 2012.

[3] L. Lignarolo, D. Ragni, C. Krishnaswami, Q. Chen, C. S. Ferreira, and G. Van Bussel. Experimental analysis of the wake of a horizontal-axis wind-turbine model. *Renewable Energy*, 70:31–46, 2014.

[4] D. Marten. *QBlade: a modern tool for the aeroelastic simulation of wind turbines.* PhD thesis, Technische Universität Berlin, 2020.

[5] D. van den Berg, D. De Tavernier, J. Gutknecht, A. Viré, and J. W. van Wingerden. The influence of floating turbine dynamics on the helix wake mixing method. *Journal of Physics: Conference Series*, 2767(3):032012, jun 2024.

[6] D. van den Berg, D. De Tavernier, D. Marten, J. Saverin, and J. van Wingerden. Wake mixing control for floating wind farms: Analysis of the implementation of the helix wake mixing strategy on the IEA 15-MW floating wind turbine. *IEEE Control Systems*, 44(5):81–105, 2024.

[7] D. van den Berg, D. De Tavernier, and J. W. van Wingerden. Using the Helix Mixing Approach on Floating Offshore Wind Turbines. *Journal of Physics: Conference Series*, 2265:042011, 2022.

[8] D. van der Hoek, B. V. den Abbeele, C. Simao Ferreira, and J. W. van Wingerden. Maximizing wind farm power output with the helix approach: Experimental validation and wake analysis using tomographic particle image velocimetry. *Wind Energy*, 2024.